# AF-SRNet: Quantitative Precipitation Forecasting Model Based on Attention Fusion Mechanism and Residual Spatiotemporal Feature Extraction

**Liangchao Geng [1,*] , Huantong Geng [1], Jinzhong Min [1], Xiaoran Zhuang [2] and Yu Zheng [3]**

1   School of Computer and Software, Nanjing University of Information Science and Technology, Nanjing 210044, China
2   Jiangsu Meteorological Observatory, Nanjing 210008, China
3   CMA Key Laboratory of Transportation Meteorology, Nanjing Joint Institute for Atmospheric Sciences, Nanjing 210041, China
*   Correspondence: 20201220015@nuist.edu.cn

**Abstract:** Reliable quantitative precipitation forecasting is essential to society. At present, quantitative precipitation forecasting based on weather radar represents an urgently needed, yet rather challenging. However, because the Z-R relation between radar and rainfall has several parameters in different areas, and because rainfall varies with seasons, traditional methods cannot capture high-resolution spatiotemporal features. Therefore, we propose an attention fusion spatiotemporal residual network (AF-SRNet) to forecast rainfall precisely for the weak continuity of convective precipitation. Specifically, the spatiotemporal residual network is designed to extract the deep spatiotemporal features of radar echo and precipitation data. Then, we combine the radar echo feature and precipitation feature as the input of the decoder through the attention fusion block; after that, the decoder forecasts the rainfall for the next two hours. We train and evaluate our approaches on the historical data from the Jiangsu Meteorological Observatory. The experimental results show that AF-SRNet can effectively utilize multiple inputs and provides more precise nowcasting of convective precipitation.

**Keywords:** quantitative precipitation forecasting; attention mechanism; multimodal fusion; spatiotemporal prediction

## 1. Introduction

Every year, extreme heavy precipitation causes serious disasters in urban areas, which seriously threatens the safety of people's lives and property. Such intense precipitation is highly heterogenous spatially and temporally. Therefore, the meteorological department has an important responsibility to study the characteristics of intense rain and carry out forecasts for disaster prevention.

The study of precipitation involves many fields such as hydrology, physics, and atmospheric circulation. High-resolution, accurate, real-time quantitative precipitation forecasting (QPF) is especially useful for preventing flood disasters and reducing socioeconomic impacts [1]. However, the characteristics of convective precipitation, such as rapid development, a short life cycle, and highly nonlinear dynamics make it challenging for prediction. According to the forecast period, precipitation forecasts can be divided into nowcasting (0–2 h) [2], short-term forecast (0–6 h) [3], short-range forecast (0–72 h) [4], medium-range forecast (3–15 days) [5], and long-range forecast (10–15 days) [6]. In general, for short-range and medium-term range forecasts, the numerical weather prediction (NWP) models provide superior predictions, but models have poor performance in nowcasting [7]. For precipitation nowcasting, meteorological radars provide precipitation observations with much higher resolutions than rain gauge networks, and there is a correlation between

the distribution and intensity of radar echoes and the precipitation rate [8]. Therefore, radar-based quantitative precipitation forecasting [9] can obtain more detailed spatial structure and temporal evolution characteristics of precipitation, and has become a research hotspot.

Precipitation nowcasting needs to extract highly nonstationary features and predicts precipitation's intensity, distribution, movement, and evolution in the coming hours. At present, radar echo extrapolation technology is currently a popular technology of precipitation nowcasting. Traditional optical flow methods [10] calculate the optical flow of consecutive radar maps under the assumption that consecutive frames will not change rapidly. However, the assumption may not hold when radar echo has a complex evolution [11]. Still, in order to predict precipitation, the precipitation should be retrieved according to the Z-R relationship [12] after the radar echo extrapolation. Therefore, the first step is to achieve radar echo extrapolation, and the second step is to convert radar reflectivity into rainfall rates through the Z–R relationship, but predicting the precipitation in the two steps will easily cause the superposition of errors and reduce the accuracy of the precipitation nowcasting. Over the past few years, deep learning techniques have been increasingly applied in quantitative precipitation forecasting. Wang et al. [13] proposed Eidetic 3D long short-term memory (E3DLSTM), which replaces the forget gate with the recall gate structure. Specifically, the forget gate determines whether past information can be "forgotten" like standard LSTMs. The recall gate uses an attentive module to compute the relationship between the encoded local patterns and the whole memory space. Wang et al. [14] proposed a spatiotemporal prediction model called PredRNN, which adds spatiotemporal memory units and connects them through a zigzag structure to integrate temporal and spatial features. By applying differencing operations on the nonstationary and approximately stationary properties in spatiotemporal dynamics, Wang et al. [15] proposed memory in memory (MIM) networks to capture complex nonstationary features in radar echo extrapolation. To alleviate the blurring and unrealistic issues for radar echo extrapolation, Geng et al. [16] proposed enforcement of the idea of the generative adversarial network and developed a generative adversarial network-residual convolution LSTM (GAN-rcLSTM) method. For short-term QPF, radar echo extrapolation remains a powerful method because of the high temporal and spatial resolutions of radar echo maps. However, these radar extrapolation-based QPF techniques suffer from the problem of uncertainty in converting radar reflectivity to rainfall amount, and thus are still limited in improving the accuracy of the QPF.

Direct use of precipitation data as input to predict rainfall within two hours is also a method by which to achieve precipitation nowcasting. Kevin Trebing et al. [17] proposed the small attention UNet (SmaAt-UNet) model, which uses the attention modules and depthwise-separable convolutions (DSC) to extract spatial features in the process of precipitation development. Song et al. [18] present a self-attention residual UNet (SE-ResUNet) model, which uses UNet as the backbone network and adds residual structure to extract spatiotemporal information. Cong et al. [19] introduced a new framework for precipitation nowcasting named Rainformer. In this work, they utilized the global features extraction unit and the gate fusion unit (GFU) in order to extract features. These methods use precipitation data as the only predictor. Directly using precipitation maps can avoid the uncertainty in converting radar reflectivity to rainfall amounts through a Z-R relationship. However, due to the sparse distribution of ground observation stations, it is difficult to achieve precise precipitation nowcasting.

In addition, ground-based radars are efficient tools for observing precipitation and its microphysical structure. Some researchers consider taking multi-source meteorological data as input to forecast the precipitation. Zhang et al. [20] proposed a dual-encoder recurrent neural network called RN-Net. It takes the rainfall data of automatic weather stations and radar-echo data as input to predict rainfall for the next 2 h. Wu et al. [21] used echo-top height and hourly rainfall datasets to establish a new dynamical Z-R relationship and achieve the radar-based quantitative precipitation estimation (RQPE) [22]. In fact, utilizing multiple variables such as radar reflectivity and precipitation rate can capture

richer physical information in QPF. However, many methods cannot realize the effective fusion of multiple input variables.

This paper proposed an attention fusion spatiotemporal residual network (AF-SRNet). Inspired by multimodal fusion and spatiotemporal prediction (MFSP-Net) [23] and squeeze-and-excitation (SE) blocks [24], we investigate this model, which includes spatiotemporal residual network and attention fusion block. The spatiotemporal residual network extracts the spatiotemporal information from radar data and precipitation data independently, and then the attention fusion block combines spatiotemporal information at the highest semantic level.

## 2. Related Work

According to the first section, radar-based quantitative precipitation forecasting (RQPF) has been widely used in precipitation nowcasting in recent years due to the spatiotemporal discontinuity of precipitation station data. The process of RQPF is shown in Figure 1. At first, station rainfall grid data is obtained after interpolation, and radar mosaic grid data is obtained after quality control; then, these two data points are input into the spatiotemporal sequence forecast model to predict precipitation in the future. Spatiotemporal feature extraction and feature fusion are important parts of the spatiotemporal sequence forecast model. However, there are some weaknesses in these two parts.

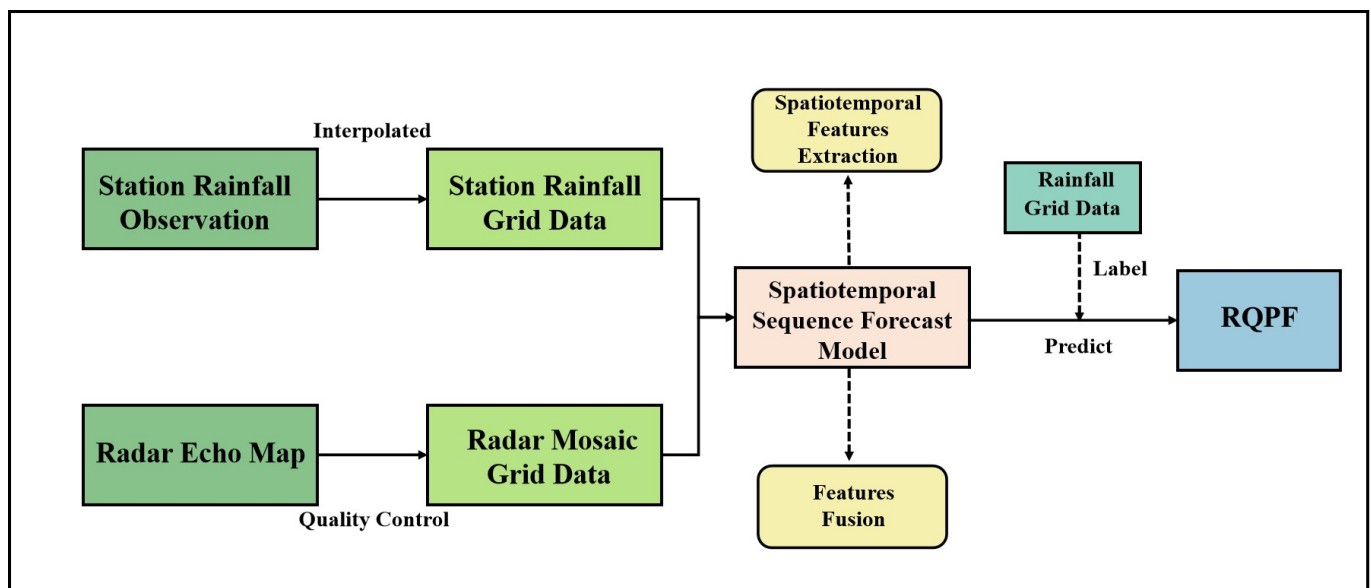

**Figure 1.** The process of radar quantitative precipitation forcasting (RQPF).

Shi et al. [25] modeled it as a spatiotemporal sequence forecasting problem, introducing the encoding-forecasting structure of ConvLSTM to achieve quantitative precipitation forecasting. Luo et al. [26] introduced a sequence-to-sequence architecture called PFST-LSTM for RQPF. However, these works can only predict radar echoes. After the radar echo extrapolation, the predicted radar echo intensity needs to be converted into rainfall rates relying on the Z–R relationship, but different regions and different scales of precipitation systems have different Z-R relationships, which causes errors in precipitation nowcasting.

Subsequently, Bouget et al. [27] fused radar echo images and wind velocity to predict precipitation, and they also directly used rainfall as the target to enhance the effect of QPF. Zhou et al. [28] proposed a model called LightningNet to achieve lightning nowcasting by combining multisource observation data at different channels. However, these methods cannot effectively fuse the spatiotemporal information of multisource data by simple summation or channel concatenation.

In addition, more and more spatiotemporal sequence forecasting models are applied in prediction tasks. Wang et al. [29] proposed the PredRNN++ with the structure of causal

LSTM and highway units to capture spatiotemporal features. Chai et al. proposed CMS-LSTM to capture multi-scale spatiotemporal flows. However, the precipitation system concludes with a more complex spatiotemporal motion, and the spatial and temporal information will affect each other in these methods.

Overall, most of the previous work for RQPF has some deficiencies. First, when extracting the spatiotemporal information from the precipitation system, some features can be lost because the temporal and spatial information will affect each other, making it very difficult to achieve precise nowcasting. Secondly, radar and precipitation have not been effectively fused, and it is difficult to extract microphysical features, resulting in underestimation of high-intensity precipitation areas.

With regard to the above problems and the improvement of the QPF quality, this research includes two important techniques in deep learning, namely encoder–decoder [30] and attention mechanism [31]. Methods based on encoder–decoder were developed for natural language processing but are widely used in spatiotemporal sequence forecasting. Attention mechanisms can adaptively learn to reassign the importance of variable features, and have been proven to perform well in fusing features. We use the structure of encoder–decoder to improve spatiotemporal extraction and the attention mechanism to improve the effect of feature fusion.

## 3. Methods

### 3.1. Problem Definition

We define the precipitation nowcasting problem as a sequence-to-sequence problem. Given the radar echo sequence data and precipitation grid sequence data in the past period, we predict the precipitation in the future period. More specifically, $R = \{R_{t-n}, R_{t-n+1}, \ldots R_t\}$ is a collection of $N$ radar echo maps from time $t - n$ to time $t$, $P = \{P_{t-n}, and\ P_{t-n+1}, \ldots P_t\}$ is the ground precipitation grid data with the same spatial and temporal resolution as the radar echo data from time $t - n$ to time $t$. The whole prediction process can be defined as follows,

$$\hat{P} = \Gamma(R, P), \tag{1}$$

where $\Gamma$ is the nowcasting model, and $\hat{P} = \{\hat{P_{t+1}}, \hat{P_{t+2}} \ldots \hat{P_{t+m}}\}$ is the predict precipitation from time $t + 1$ to time $t + m$.

### 3.2. Model

#### 3.2.1. Whole Network

The overall network architecture of AF-SRNet is shown in Figure 2. Inspired by LightNet [32] and STRPM [33], AF-SRNet consists of two encoders and a decoder. Encoders have four spatiotemporal residual units (SRUs). The radar encoder extracts spatiotemporal features from radar echo maps. The precipitation encoder extracts spatiotemporal features of precipitation maps so that the two will not interfere in the early stage. The fusion module based on the attention mechanism fuses the spatiotemporal features extracted by the two encoders. Finally, the fused features are input to the precipitation decoder and predict future rainfall. The operation and the aim of each part are now detailed.

#### 3.2.2. Spatiotemporal Residual Unit

Some methods use the spatiotemporal long short-term memory unit (STLSTM) to extract spatiotemporal information. However, the temporal and spatial data will affect each other, making it difficult to extract the complex motion features in the precipitation evolution. To deal with this problem, we designed a spatiotemporal residual unit (SRU) to focus on modeling temporal evolution information and spatial evolution information between previous and future frames in the feature space, which is shown in Figure 3. The SRU consists of three modules: a temporal module, a spatial module, and a residual spatiotemporal module. Each module includes a structure of residual. They can effectively utilize the previous spatiotemporal state information so that the feature extraction has a wider spatiotemporal receptive field.

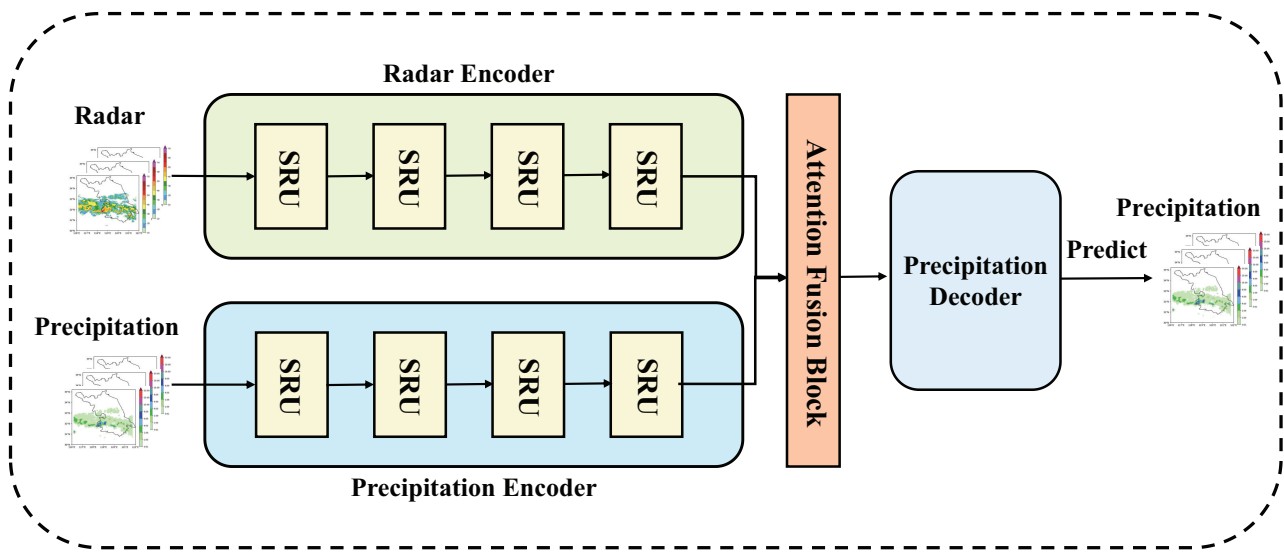

**Figure 2.** Architecture of AF-SRNet. It contains two encoders, a decoder, and a fusion block.

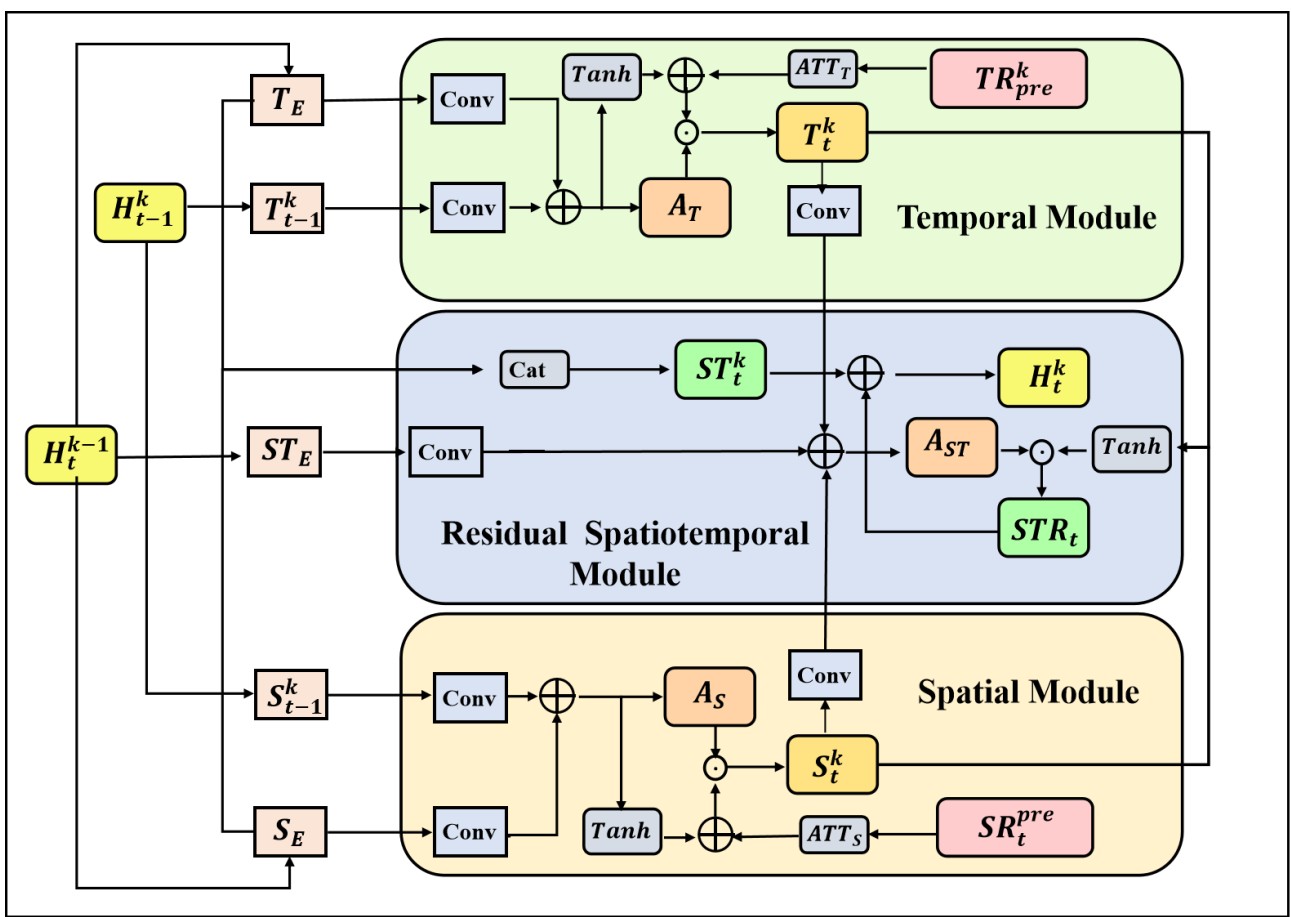

**Figure 3.** The structure of SRU. The SRU includes a temporal module, a spatial module, and a residual spatiotemporal module. The temporal module extracts time-series information, the spatial module extracts spatial evolution features, and the residual spatiotemporal module fuses spatial and temporal information.

To preserve more useful temporal features from the past and use the current temporal features to capture long-term and short-term dependencies, the temporal module jointly

utilizes multiple temporal states to obtain the output state $H_t^k$, and the calculation formula can be expressed as follows,

$$T_t^k = \left( \tanh\left( W_{T_E} T_E + W_{T_{t-1}^k} T_{t-1}^k \right) + ATT_T\left( TR_{pre}^k \right) \right) \odot A_T$$
$$A_T = \sigma\left( W_{T_E} T_E + W_{T_{t-1}^k} T_{t-1}^k \right),$$

(2)

where $A_T$ denotes the temporal residual gate, which can model the interframe residual temporal information, $\tanh\left( W_{T_E} T_E + W_{T_{t-1}^k} T_{t-1}^k \right)$ represents the current temporal information, and $ATT_T\left( TR_{pre}^k \right)$ represents the preserved temporal information from previous time steps before $t$. In particular, $ATT_T(\cdot)$ denotes the temporal attention network which is constructed with convolutional layers and can help fuse the multiple temporal states according to the degree of importance. In order to effectively utilize the information of multiple spatial states and capture global features and local features, similar to the temporal module, the spatial module jointly utilizes multiple spatial states to obtain the output spatial state $S_t^k$, the state-to-state transitions can be represented as follows,

$$S_t^k = \left( \tanh\left( W_{S_E} S_E + W_{S_{t-1}^k} S_{t-1}^k \right) + ATT_S\left( SR_t^{pre} \right) \right) \odot A_S$$
$$A_T = \sigma\left( W_{S_E} S_E + W_{S_{t-1}^k} S_{t-1}^k \right),$$

(3)

where $A_S$ denotes the spatial residual gate, which can model the interframe residual spatial information, $\tanh\left( W_{S_E} S_E + W_{S_{t-1}^k} S_{t-1}^k \right)$ represents the spatial information of current layer, and $ATT_S\left( SR_t^{pre} \right)$ represents the spatial information from previous layers. In particular, $ATT_S(\cdot)$ denotes the spatial attention network which is constructed with convolutional layers and can help fuse the multiple spatial states according to the degree of importance.

The residual spatiotemporal module aggregates all spatiotemporal information to the final hidden state $H_t^k$. The following are the calculation equations,

$$A_{ST} = \sigma\left( W_{ST_{S_E}} ST_{S_E} + W_{T_t^k} T_t^k + W_{S_t^k} S_t^k \right)$$
$$ST_t^k = W_{1 \times 1} * [\, T_E, S_E]$$
$$STR_t = A_{ST} \odot \tanh\left( W_{1 \times 1} * \left[ T_t^k, S_t^k \right] \right)$$
$$H_t^k = S\, T_t^k + STR_t,$$

(4)

where $A_{ST}$ denotes the residual gate, which is utilized to aggregate the predicted temporal and spatial residual information. $ST_t^k$ represents the spatiotemporal input features and $STR_t$ represents the spatiotemporal residual features between previous and future frames.

To further extract more efficient deep spatiotemporal features, four SRUs are typically stacked into a single encoder, as shown in Figure 4. For the SRU at time step $t$ in layer $k$, the temporal features $T_E$, the spatial features $S_E$, and spatiotemporal features $ST_E$ are fed into the corresponding modules of SRU. In particular, for $k > 1$, these features are represented with the hidden state from the previous layer $H_t^{k-1}$. The hidden state of the last time step $H_{t-1}^k$ includes the spatial state $S_{t-1}^k$ and the temporal state $T_{t-1}^k$; they are fed into the temporal module and spatial module, respectively. In addition, the input of the SRU also includes the temporal residual information extracted at the time before t $TR_{pre}^k$ and the spatial residual information before the $k$ layer $SR_{pre}^k$. In particular, when $k = 1$, there is no spatial residual information.

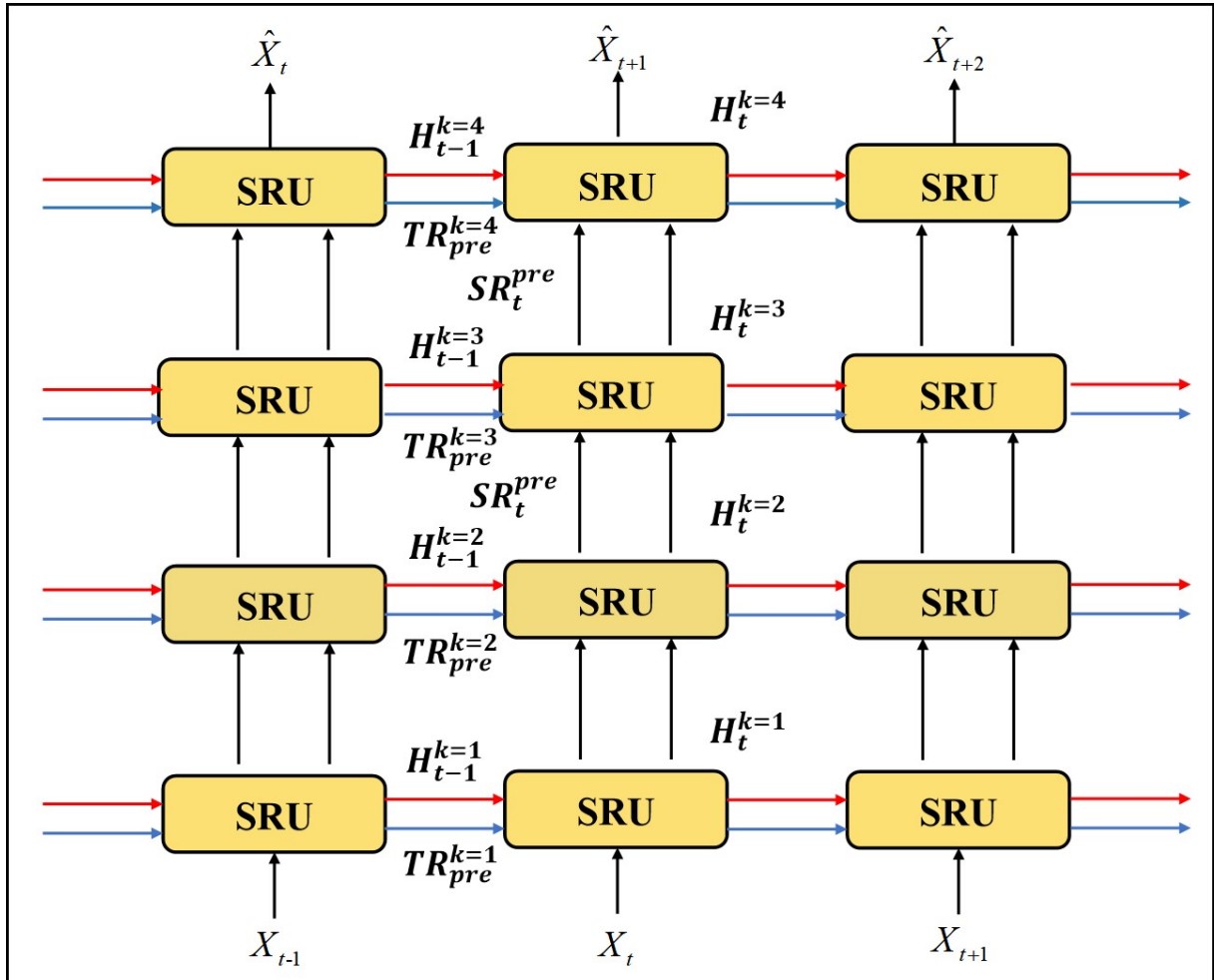

**Figure 4.** Stacked SRU structure.

### 3.2.3. Attention Fusion Block

In deep learning models, a typical method of using multiple input variables is to concatenate the variable fields at the "channel" dimension and put them into the first convolutional layer [34]. This fusion method is also called early fusion [28]. However, more complex dependencies exist between input variables, incurring possible information entanglement effects. We adopt the late-fusion strategy [35] by using two encoders to independently extract radar echo features and precipitation features. Then two final hidden states at the highest semantic level $H_{radar}$ and $H_{precip}$ are obtained. Then we use a multi-scale attention mechanism as shown in Figure 5. AFB constrains the local and global features through the attention mechanism, which greatly alleviates the numerical difference and avoids the problem that linear fusion cannot play a role due to the significant difference between them. Moreover, AFB can effectively leverage the microphysics feature of the precipitation system and achieve more precise precipitation nowcasting.

The fusion module based on the multi-scale attention mechanism takes the radar echo hidden state $H_{radar}$ and the precipitation hidden state $H_{precip}$ as input, obtaining the fused state $H_{fusion}$ through the pointwise convolution (PWConv), and then $H_{fusion}$ is input into the precipitation decoder for precipitation prediction. The entire attention fusion block can be expressed as:

$$H_{fusion} = M(H_{radar} \oplus H_{precip}) \otimes H_{radar} + (1 - M(H_{radar} \oplus H_{precip})) \otimes H_{precip}$$
$$M(X) = \sigma(L(X) \oplus G(X)),$$

(5)

where $M(\cdot)$ denotes the attentional weights, which is a parameter obtained through network learning, $L(\cdot)$ represents the convolution operation of local features, and $G(\cdot)$ represents the convolution operation of global features. The specific implementation is shown in Figure 3, where $\oplus$ denotes the broadcasting addition and $\otimes$ denotes the element-wise multiplication.

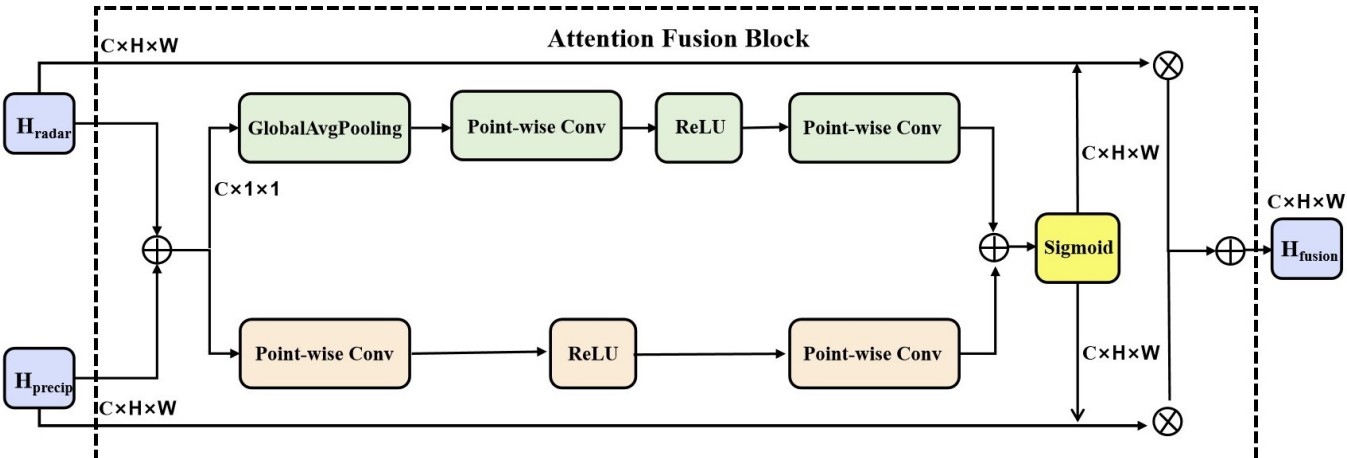

**Figure 5.** The structure of attention fusion block. By using the attention mechanism, the radar echo hidden state $H_{radar}$ and the precipitation hidden state $H_{precip}$ extracted by encoders are fused to obtain the fusion state $H_{fusion}$.

### 3.2.4. Decoder

Similar to the encoders, we utilize the multiple spatiotemporal decoders to decode the fused features from low-dimensional feature space back to high-dimensional temporal and spatial data space, respectively, and then we can predict the subsequent frames.

## 4. Experiments

### 4.1. Dataset

This study uses the radar echo data and gridded precipitation observations from April to September in 2019–2021 in Jiangsu Province, China.

Radar reflectivity dataset: This dataset is the time series of radar echo data, the physical meaning of which is the radar-based reflectivity at the height of 3 km. The higher the concentration of water droplets in the atmosphere, the higher the radar reflectivity. This dataset is obtained after quality control and networking of several S-band weather radars in Jiangsu, covering the entire area of Jiangsu Province. The data value range is 0–70 dBZ, the horizontal resolution is 0.01° (about 1 km), the time resolution is 6 min, and the grid size of single-time data is 480 × 560 pixels.

Precipitation dataset: This dataset is obtained by interpolating the precipitation data of automatic meteorological observation stations in Jiangsu to a uniform grid through the Cressman interpolation method [36]. Moreover, precipitation is the accumulated precipitation of the automatic station in 6 min; that is, the accumulated value of the precipitation observation in 6 min up to the current time. The value range is 0–10 mm. The horizontal resolution, the time resolution, and the horizontal size are the same as the radar.

In terms of data preprocessing, we first downsampled the original resolution data to a size of 120 × 140 pixels through max pooling, considering the limitations of computational costs. The horizontal resolution after downsampling is 0.04° (approximately 4 km) considering the limitations of computing power and training costs. We downsampled the original resolution data to a size of 120 × 140 pixels, and the horizontal resolution after downsampling is 0.04°(approxmately 4 km), as shown in Figure 6. It can be seen from the figure that there is a good correspondence between the high radar echo area and the heavy precipitation area. Secondly, in order to predict the precipitation in the next

two hours, we determined that we should use the past 20 times (2 h) data to predict the precipitation of the next 20 times (2 h). The data was divided into 5143 groups; each group included 40 consecutive frames. The first 20 frames are used as the model's input, and the last 20 frames were used as the ground truth. After that, we divided these sequence data into the training set, validation set, and test set according to the ratio of 8:1:1. Finally, we normalized both the precipitation data and the radar data to a range of 0–1.0.

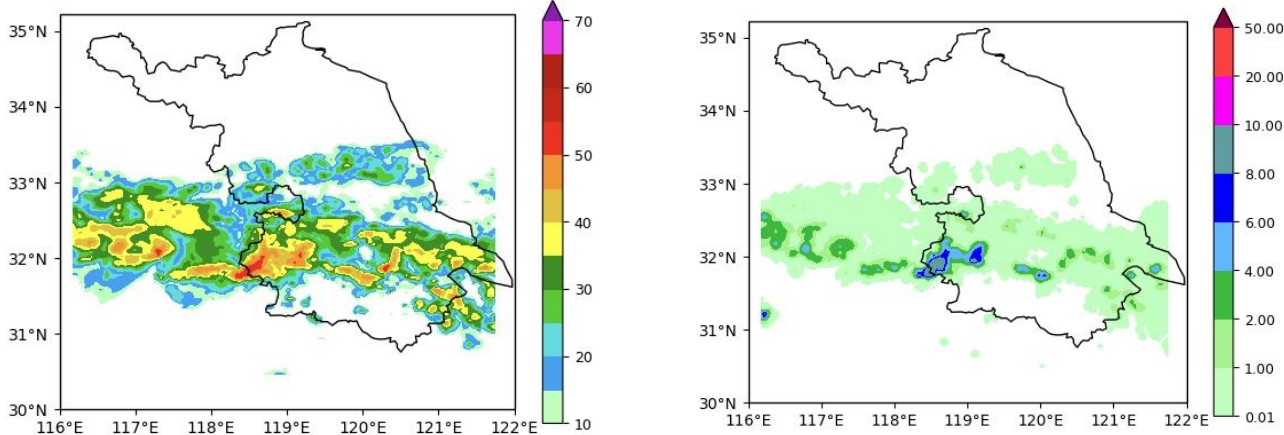

**Figure 6.** Visual display of processed radar echo and precipitation data.

### 4.2. Loss Fuction

The statistical results of precipitation distribution according to different rainfall intensities are shown in Figure 7. It can be seen that there exists the problem of imbalanced frequencies of different rainfall levels in precipitation data. Specifically, among these categories, rainfall above 2 mm is the lowest proportion with a percentage of 2.2%, and rainfall between 0 mm to 0.2 mm is larger than rainfall above 2 mm.

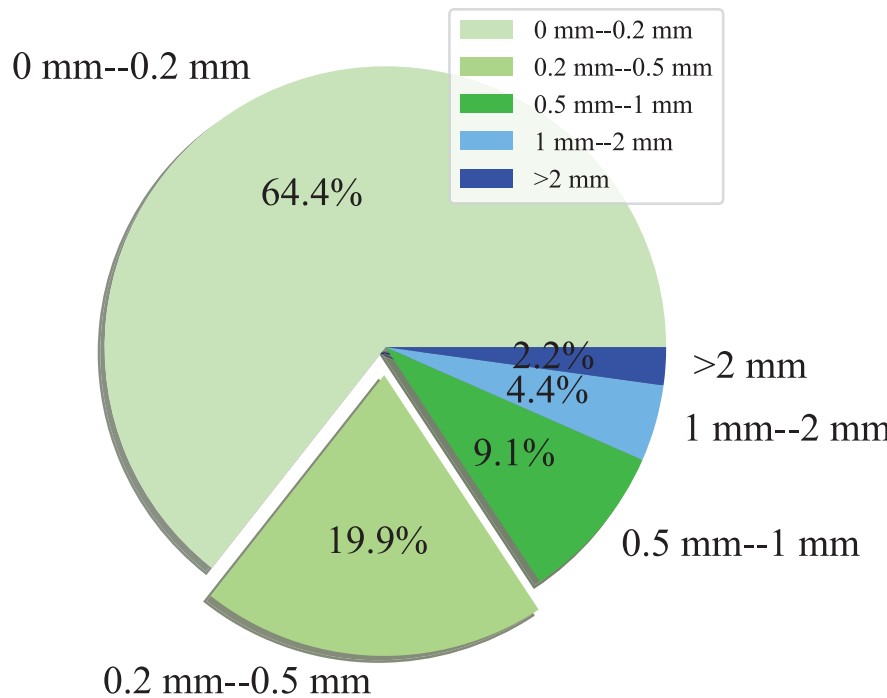

**Figure 7.** Precipitation statistics in the dataset.

Therefore, according to [37], we adopt a weighted mean absolute error (WMAE) loss scheme. The loss function is defined as follows:

$$\text{Loss} = 1/(20 * 120 * 140) \sum_{t}^{20} \sum_{j}^{120} \sum_{k}^{140} W(Y_t[j,k]) * \left| Y_t[j,k] - \hat{Y}_t[j,k] \right|$$

$$W[x] = \begin{cases} 1 & x < 0.2 \\ 2.5 & 0.2 <= x < 0.5 \\ 5 & 0.5 <= x < 1 \\ 10 & 1 <= x < 2 \\ 20 & x > 2 \end{cases}, \tag{6}$$

where $Y_t$ represent the actual $t$th six minutes accumulated rainfall and $\hat{Y}_t$ represent the predicted version.

### 4.3. Implementation Details

PyTorch [38] implements all models in this paper with a NVIDIA RTX A100 GPU. All models were trained by using the Adam optimizer [39] with a starting learning rate of $10^{-4}$. In addition, to ensure that experimental results are comparable, all models had the same hyperparameters. All described models were trained for a maximum of 30 epochs, and we also used an early stopping strategy when the validation loss did not increase. The batch size was set to 8.

### 4.4. Performance Metric

In order to evaluate the performance of our model quantitatively, we use multiple metrics from the meteorological field. Because meteorologists are more concerned about the model performance under different rainfall levels, we binarize our prediction and the ground truth with different thresholds. If the value is larger than the given threshold, we set the corresponding value to 1; otherwise we set it to 0. Then, we calculate the number of positive predictions TP (truth = 1, prediction = 1), false-positive predictions FP (truth = 0, prediction = 1), true negative predictions TN (truth = 0, prediction = 0) and false negative predictions FN (truth = 1, predition = 0). At last, the critical success index (CSI), probability of detection (POD), false alarm rate (FAR) and Heidke skill sore (HSS) [40] can be computed as follows:

$$\begin{aligned} \text{CSI} &= \frac{\text{TP}}{\text{TP} + \text{FN} + \text{FP}} \\ \text{POD} &= \frac{\text{TP}}{\text{TP} + \text{FN}} \\ \text{FAR} &= \frac{\text{FP}}{\text{TP} + \text{FP}} \\ \text{HSS} &= \frac{\text{TP} \times \text{TN} - \text{FN} \times \text{FP}}{(\text{TP} + \text{FN})(\text{FN} + \text{TN}) + (\text{TP} + \text{FP})(\text{FP} + \text{TN})}. \end{aligned} \tag{7}$$

Note that for CSI, POD, and HSS, the larger the better, whereas for FAR, the smaller the better.

Generally speaking, precipitation is divided into five categories: light rain, moderate rain, heavy rain, rainstorm, and downpour. As shown in Table 1, we classified 6-min rainfall into five different grades according to the study mentioned in [41].

For 1-h rainfall, according to [23], precipitation is divided into four categories, as shown in Table 2, so we choose 0.5 mm/h, 2 mm/h, and 5 mm/h as the thresholds for 1-h precipitation evaluation.

**Table 1.** Categories of 6-min rainfall.

| Category | 6-min Rainfall (mm) |
|---|---|
| Drizzle | [0, 0.1) |
| Light/moderate rain | [0.1, 0.7) |
| Heavy rain | [0.7, 1.5) |
| Rainstorm | [1.5, 4) |
| Downpour | [4, ∝) |

**Table 2.** Categories of 1-h rainfall.

| Rainfall Levels | Rainfall Amount per Hour (mm) |
|---|---|
| No or hardly noticeable | [0, 0.5) |
| Light | [0.5, 2) |
| Light to moderate | [2, 5) |
| Moderate or greater | [5, ∝) |

*4.5. Experimental Results and Comparisons with SOTAs*

We use some spatiotemporal prediction models as the benchmark models for precipitation nowcasting, including ConvLSTM, PredRNN, Memory In Memory, and SE-ResUNet. In order to ensure the comparability of the experiments, all models utilize radar and precipitation data, and we concatenate fields at the "channel" dimension of the two data as the input of these models. Each model uses the past 20 times as input and predicts the following 20 times in the future; that is, it predicts the rainfall in the next 0–2 h. When the validation loss no longer decreases during the training phase, the model with the smallest validation loss is selected as model well trained for prediction. Due to the low rainfall amounts of 6 min, we calculated the cumulative rainfall for an hour to evaluate the performance of these models. The average evaluation results of one frame of precipitation amount nowcasting in the first hour are shown in Table 3, and the average evaluation results of two frames of precipitation amount nowcasting in the first two hours are shown in Table 4.

**Table 3.** Average evaluation results of one frame of precipitation amount nowcasting in the first hour. The best performance is highlighted in bold. "↑" means that the higher the score, the better, while "↓" means that the lower the score, the better.

| Method | r ≥ 0.5 mm/h | | | | r ≥ 2.0 mm/h | | | | r ≥ 5.0 mm/h | | | |
|---|---|---|---|---|---|---|---|---|---|---|---|---|
| | CSI↑ | POD↑ | FAR↓ | HSS↑ | CSI↑ | POD↑ | FAR↓ | HSS↑ | CSI↑ | POD↑ | FAR↓ | HSS↑ |
| ConvLSTM | 0.4169 | 0.4548 | 0.1570 | 0.2528 | 0.2767 | 0.3285 | **0.2365** | 0.1802 | 0.1210 | 0.1522 | 0.2477 | 0.0865 |
| PredRNN | 0.4140 | 0.4545 | 0.1549 | 0.2517 | 0.2740 | 0.3316 | 0.2758 | 0.1798 | 0.1254 | 0.1623 | 0.2719 | 0.0895 |
| MIM | 0.4328 | 0.4651 | **0.1323** | 0.2629 | 0.2847 | 0.3314 | 0.2534 | 0.1870 | 0.1182 | 0.1387 | **0.2124** | 0.0839 |
| SE-ResUNet | 0.4168 | 0.5536 | 0.3327 | 0.2496 | 0.2619 | 0.4248 | 0.4712 | 0.1720 | 0.1272 | 0.2427 | 0.4951 | 0.0919 |
| AF-SRNet | **0.5159** | **0.6511** | 0.3051 | **0.3071** | **0.3360** | **0.2499** | 0.4643 | **0.2178** | **0.1545** | **0.2499** | 0.4274 | **0.1073** |

As shown in the above tables, we can see that AF-SRNet performs best in almost all metrics. This means that the SRNet proposed in this paper can effectively extract the spatiotemporal information in the precipitation system. Moreover, the model can fully use the correlation between the radar high echo area and the precipitation high-intensity area by using an attention fusion block, which improves the accuracy of short-term heavy rainfall prediction to a certain extent. Secondly, MIM performs better than ConvLSTM, the PredRNN, and the SE-ResUNet, as it can capture short-term dynamic features. As is well known, convective precipitation has the the characteristic of rapid development, and the MIM can model the non-stationary and extract complex features. Last but not least, it can be seen from Table 4 that with the increase of prediction time, the effect in SE-ResUNet is

constantly enhanced, even under the threshold of 5 mm/h. The POD is higher than our model, but the FAR is also increasing at the same time. We suspect that the SE-ResUNet model can extract spatial features well, but it is difficult to extract temporal evolution information and capture the decline process in areas of high precipitation intensity.

**Table 4.** Average evaluation results of two frames of precipitation amount nowcasting in the first two hours. The best performance is highlighted in bold. "↑" means that the higher the score, the better, while "↓" means that the lower the score, the better.

| Method | r ≥ 0.5 mm/h | | | | r ≥ 2.0 mm/h | | | | r ≥ 5.0 mm/h | | | |
|---|---|---|---|---|---|---|---|---|---|---|---|---|
| | CSI↑ | POD↑ | FAR↓ | HSS↑ | CSI↑ | POD↑ | FAR↓ | HSS↑ | CSI↑ | POD↑ | FAR↓ | HSS↑ |
| ConvLSTM | 0.3436 | 0.3845 | 0.2324 | 0.2097 | 0.2097 | 0.2580 | 0.3076 | 0.1387 | 0.0803 | 0.1052 | 0.2827 | 0.0582 |
| PredRNN | 0.3436 | 0.3867 | 0.2236 | 0.2104 | 0.2114 | 0.2637 | 0.3249 | 0.1408 | 0.0872 | 0.1151 | 0.2933 | 0.0630 |
| MIM | 0.3531 | 0.3890 | **0.1987** | 0.2165 | 0.2110 | 0.2530 | **0.3023** | 0.1412 | 0.0798 | 0.0961 | **0.2536** | 0.0574 |
| SE-ResUNet | 0.3475 | 0.5395 | 0.4724 | 0.2079 | 0.2235 | 0.3827 | 0.6118 | 0.1577 | 0.1024 | **0.2217** | 0.6819 | 0.0790 |
| AF-SRNet | **0.4196** | **0.5438** | 0.3662 | **0.2507** | **0.2560** | **0.4049** | 0.5039 | **0.1673** | **0.1121** | 0.1944 | 0.4558 | **0.0792** |

In order to further visually represent the AF-SRNet prediction ability for high-intensity rainfall, a randomly chosen of visualization example is shown in Figure 7.

The first two columns of each row are the 6-minute precipitation results, and the last two are the 1-hour precipitation results. First, Figure 8 shows that our model can predict the area and the intensity of high-intensity rainfall that is much better than other SOTAs. However, our model undeniably suffers from overestimation in some areas related to the deep fusion of radar and precipitation features. Secondly, as the prediction time increases, ConvLSTM and MIM present bad performance on predicting high-intensity precipitation. The SE-ResUNet seems to have better prediction details, but the effect of the rainfall area forecast is worse than other models. This is not surprising because these models cannot fully utilize the advantages of multi-source data through a simple fusion. In summary, the observation suggests that as the intensity of rainfall increases, the superiority of AF-SRNet becomes more obvious.

We draw Figure 9 to describe the MSE curves of all models at all nowcasting lead time stamps on the whole test set. We can see that our model has a lower prediction error in the first seven times. It is worth pointing out that except for SE-ResUNet, our model has a lower prediction error than other models' rest times, demonstrating our approach's effectiveness. Similar to the above analysis, We suspect that the UNet model can extract spatial features well but cannot extract temporal evolution information, resulting in a high false alarm rate.

### 4.6. Ablation Experiments and Analyses

In this section, we explore the impact of the spatiotemporal residual network and attention fusion block proposed in this paper by ablation experiments. Because spatiotemporal LSTM (STLSTM) is a common spatiotemporal feature extraction network, we use STLSTM as the baseline model.

The results of the AF-SRNet without attention fusion block (denoted by SRNet) and STLSTM are compared to see the impact of the spatiotemporal residues feature extracted by SRNet. As shown in Tables 5 and 6, it can be seen that SRNet outperforms STLSTM under all thresholds. The results indicate that SRNet extracts features from temporal information and spatial information, respectively, which can effectively avoid the mutual interference of temporal and spatial information.

To demonstrate the validity of attention fusion block (AF), we compare the results of AF-SRNet without AF (denoted as SRNet) with the complete AF-SRNet. AF-SRNet outperforms SRNet in all metrics. In addition, it can also be seen that when the threshold is 5 mm/h, the effect of STLSTM with AF block (denoted as AF-STLSTM) is better than

SRNet. Consequently, we can say that the AFblock proposed in this paper is superior in fusing radar echo and precipitation features.

To make a visual comparison of the four methods, we depict the prediction results on an example in Figure 10.

We can make the following conclusions from the above figure. First, the method with the AF block performs well in the prediction of high-intensity precipitation, which indicates that our method of fusing radar echoes and precipitation features is effective. Secondly, SRNet performs significantly better than STLSTM, which indicates that the separate modeling strategy of temporal and spatial features proposed in this paper can extract more complex motion information. Finally, combining SRNet with AF block can achieve more accurate precipitation forecasts.

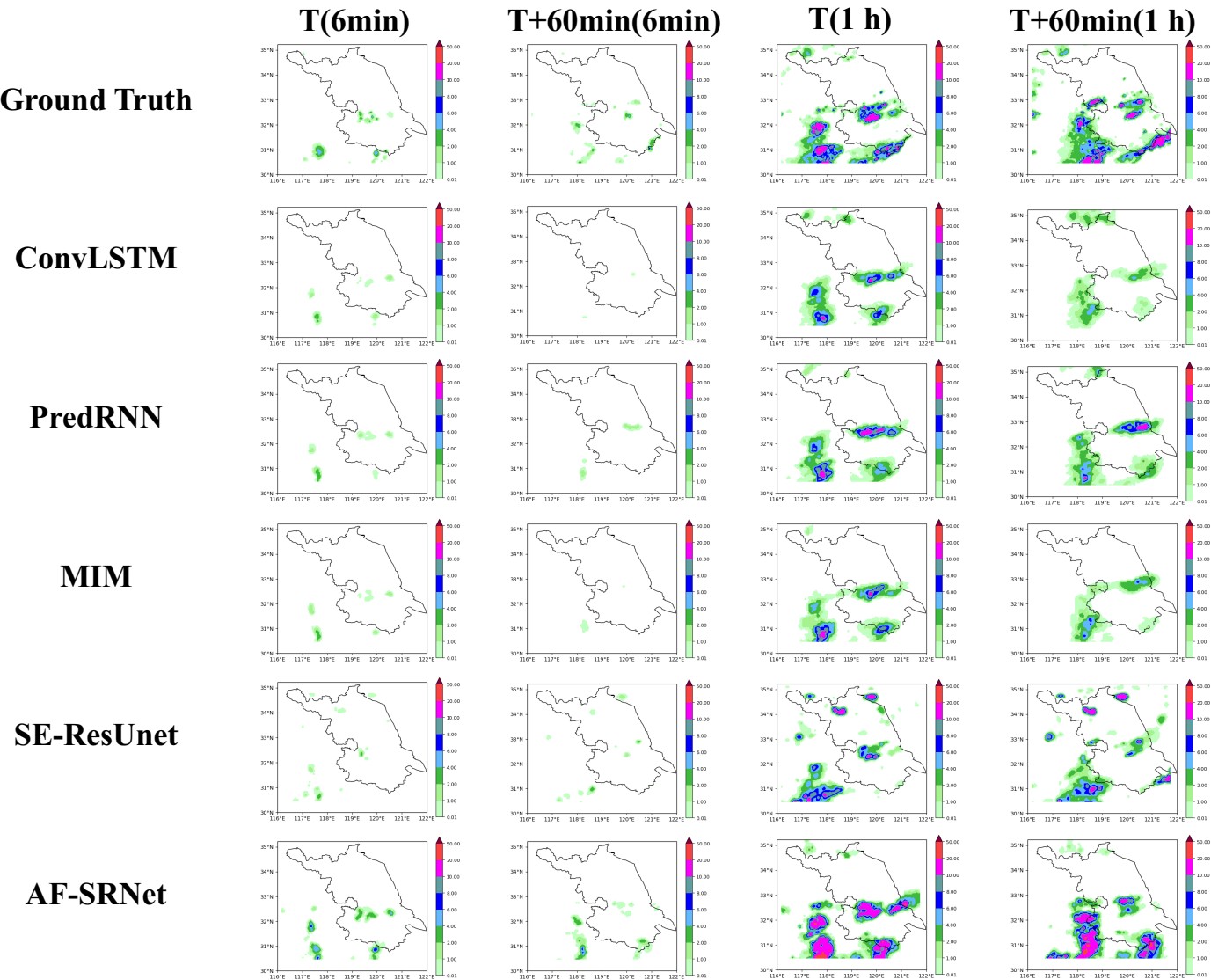

**Figure 8.** Visual comparison with other SOTAs results. The first row is the ground truth, and the last row is the effect of our model. The first column of each row is the 6-minute cumulative precipitation from time T to time T + 6 min, the second column is the 6-minute cumulative precipitation from time T + 60 min to time T + 66 min, the third column is the 1-h cumulative precipitation from time T to time T + 60 min, and the fourth column is the 1-h cumulative precipitation from time T + 60 min to time T + 120 min.

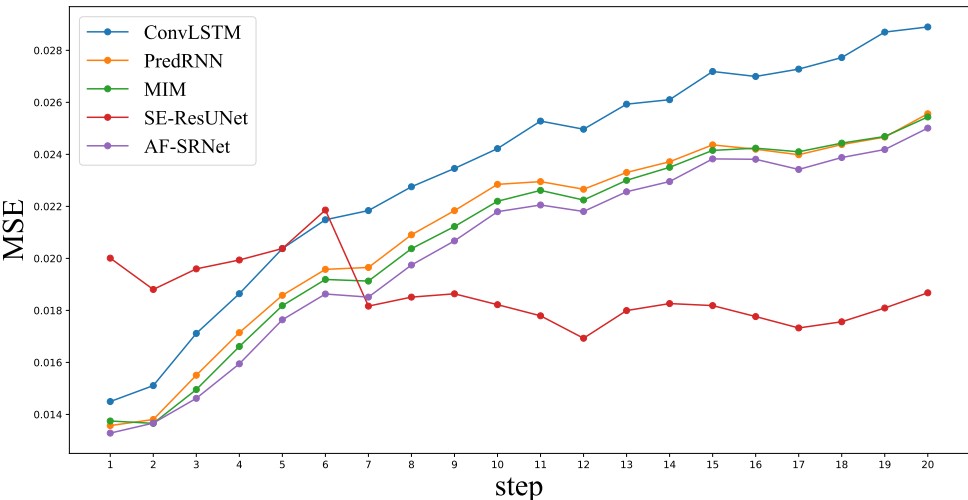

**Figure 9.** MSE comparison with different time intervals for the next 20 frames, with a 6-min interval between each frame.

**Table 5.** Average evaluation results of one frame of precipitation amount nowcasting in the first hour. The best performance is highlighted in bold. "↑" means that the higher the score, the better, while "↓" means that the lower the score, the better.

| Method | r ≥ 0.5 mm/h | | | | r ≥ 2.0 mm/h | | | | r ≥ 5.0 mm/h | | | |
|---|---|---|---|---|---|---|---|---|---|---|---|---|
| | CSI↑ | POD↑ | FAR↓ | HSS↑ | CSI↑ | POD↑ | FAR↓ | HSS↑ | CSI↑ | POD↑ | FAR↓ | HSS↑ |
| STLSTM | 0.4140 | 0.4545 | **0.1549** | 0.2517 | 0.2740 | 0.3316 | **0.2758** | 0.1798 | 0.1254 | 0.1623 | **0.2719** | 0.0895 |
| AF-STLSTM | 0.5025 | 0.6143 | 0.2892 | 0.3016 | 0.3300 | 0.4579 | 0.4296 | 0.2156 | 0.1506 | 0.2212 | 0.4123 | 0.1060 |
| SRNet | 0.4957 | 0.5970 | 0.2791 | 0.2985 | 0.3243 | 0.4439 | 0.4364 | 0.2128 | 0.1465 | 0.2156 | 0.4422 | 0.1035 |
| AF-SRNet | **0.5159** | **0.6511** | 0.3051 | **0.3071** | **0.3360** | 0.2499 | 0.4643 | **0.2178** | **0.1545** | 0.2499 | 0.4274 | **0.1073** |

**Table 6.** Average evaluation results of two frames of precipitation amount nowcasting in the first two hours. The best performance is highlighted in bold. "↑" means that the higher the score, the better, while "↓" means that the lower the score, the better.

| Method | r ≥ 0.5 mm/h | | | | r ≥ 2.0 mm/h | | | | r ≥ 5.0 mm/h | | | |
|---|---|---|---|---|---|---|---|---|---|---|---|---|
| | CSI↑ | POD↑ | FAR↓ | HSS↑ | CSI↑ | POD↑ | FAR↓ | HSS↑ | CSI↑ | POD↑ | FAR↓ | HSS↑ |
| STLSTM | 0.3436 | 0.3867 | **0.2236** | 0.2104 | 0.2114 | 0.2637 | **0.3249** | 0.1408 | 0.0872 | 0.1151 | **0.2933** | 0.0630 |
| AF-STLSTM | 0.4113 | 0.5192 | 0.3794 | 0.2485 | 0.2532 | 0.3747 | 0.4903 | 0.1662 | 0.1071 | 0.1679 | 0.4587 | 0.0766 |
| SRNet | 0.3994 | 0.4935 | 0.3713 | 0.2424 | 0.2459 | 0.3541 | 0.4813 | 0.1633 | 0.1038 | 0.1623 | 0.4746 | 0.0745 |
| AF-SRNet | **0.4196** | **0.5438** | 0.3662 | **0.2507** | **0.2560** | **0.4049** | 0.5039 | **0.1673** | **0.1121** | **0.1944** | 0.4558 | **0.0792** |

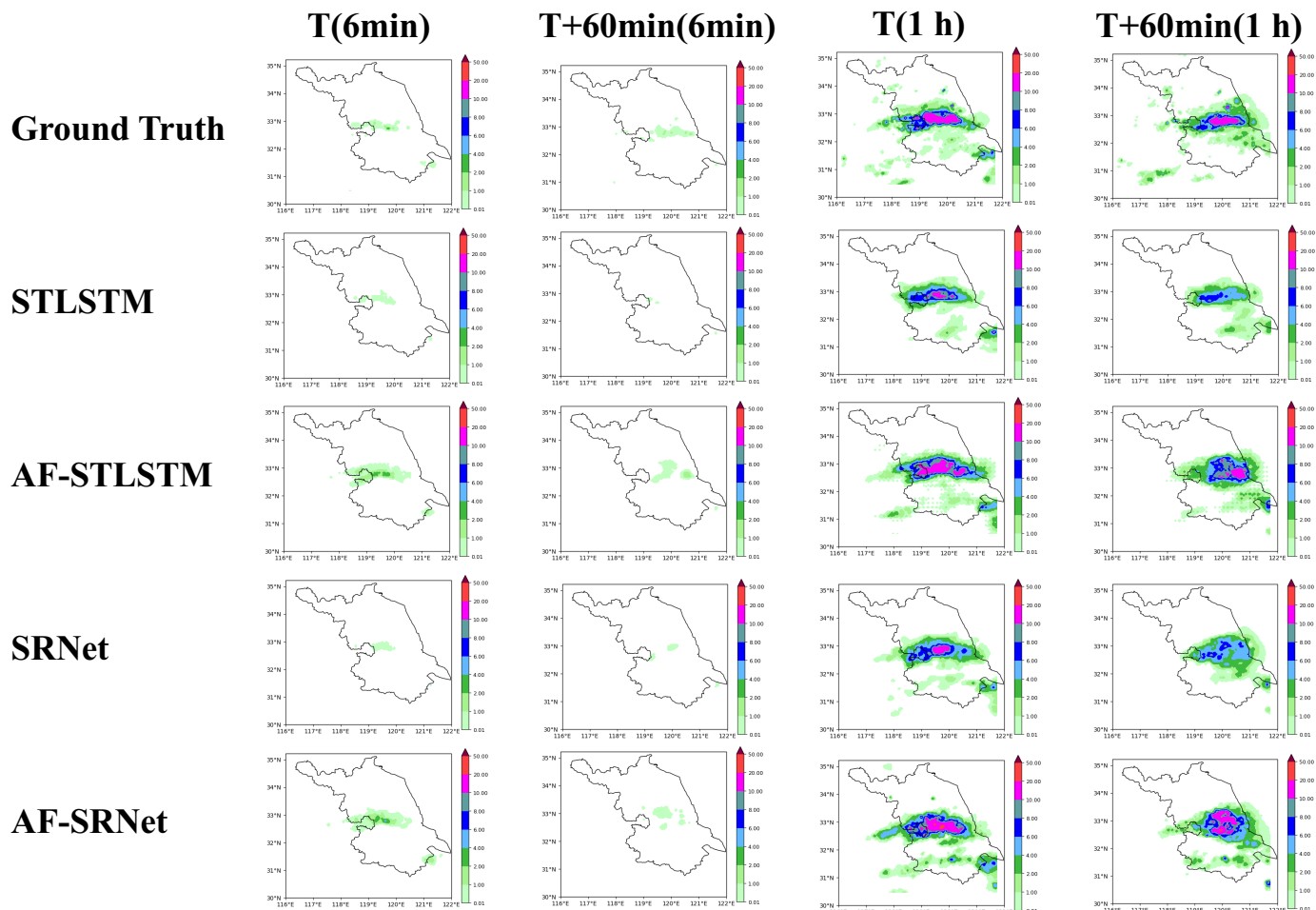

**Figure 10.** Visual comparison with ablation results. The first column of each row is the 6-min cumulative precipitation from time T to time T + 6 min, the second column is the 6-min cumulative precipitation from time T + 60 min to time T + 66 min, the third column is the 1-h cumulative precipitation from time T to time T + 60 min, and the fourth column is the 1-h cumulative precipitation from time T + 60 min to time T + 120 min.

## 5. Discussion

Currently, some precipitation nowcasting methods based on spatiotemporal sequence prediction suffer from the problem that spatial and temporal information affect each other. Moreover, most radar-based quantitative precipitation forecasting methods only use a simple fusion method to utilize radar and precipitation data. It is difficult to effectively establish microphysical constraints in developing precipitation systems. To solve these problems, we explore the combination of independent spatiotemporal modeling and multimodal fusion in precipitation nowcasting. The AF-SRNet proposed in this paper uses both radar echo data and precipitation grid data as input to predict the rainfall in the next 0–2 h. By comparing the experimental results and visualization cases, we can draw the following conclusions.

First, the precipitation grid data is obtained by interpolating station data, causing the characteristic of weak continuity. Although the radar high echo area has a good correspondence with the heavy precipitation area, we can effectively improve the quantitative precipitation forecasting effect by fusing radar and precipitation features, especially for heavy precipitation forecasting.

Secondly, the extraction of temporal and spatial evolution information plays an important role in precipitation nowcasting, which affects the area and intensity of precipitation accordingly. For this, the proposed AF-SRNet utilizes multiple residual spatiotempo-

ral encoders to get a wider spatiotemporal receptive field and establish long-term and short-term dependencies.

Thirdly, it can be seen from the experimental results that all models tend to blur with the increase of forecasting steps. Therefore, we hope to improve the details of precipitation nowcasting in our future study.

## 6. Conclusions

This paper aims at exploring making full use of multi-source meteorological variables to improve the performance of quantitative precipitation forecasting. We proposed an attention fusion spatiotemporal residual network (AF-SRNet) for radar quantitative precipitation forecasting. We design the spatiotemporal residual unit to extract deep features in the spatial and temporal domains, respectively. In addition, we design the attention fusion (AF) block for fusing radar and precipitation features and improving precipitation nowcasting. Furthermore, we also use an improved loss function (WMAE) to overcome the imbalanced distribution of dataset. Experimental results showed that the proposed model performs well in forecasting precipitation area and heavy precipitation.

Generally speaking, the field of precipitation nowcasting has immense scope for improvement. In the future, we will consider fusing more meteorological elements to establish microphysical constraints in the precipitation process. In addition, we will also explore the fusion methods to solve the problem that radar echo data excessively affects precipitation forecasting.

**Author Contributions:** Conceptualization, L.G.; methodology, L.G.; software, L.G.; validation, L.G.; formal analysis, L.G.; investigation, L.G., H.G. and J.M.; resources, H.G.; data curation, X.Z. and Y.Z.; writing—original draft preparation, L.G.; writing—review and editing, H.G. and J.M.; visualization, L.G.; supervision, H.G.; project administration, H.G.; funding acquisition, H.G. All authors have read and agreed to the published version of the manuscript.

**Funding:** This research was funded by National Key Research Development Plan under Grant 2017YFC1502104 and the Beijing foundation of NJIAS under Grant BJG202103.

**Institutional Review Board Statement:** Not applicable.

**Informed Consent Statement:** Not applicable.

**Data Availability Statement:** The data presented in this study are available on request from the corresponding author. The data are not publicly available due to the confidentiality policy of Jiangsu Meteorological Observatory.

**Conflicts of Interest:** The authors declare no conflict of interest.

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
