# Peer review of "AF-SRNet: Quantitative Precipitation Forecasting Model Based on Attention Fusion Mechanism and Residual Spatiotemporal Feature Extraction"

_remotesensing, doi:10.3390/rs14205106_

Round 1

Reviewer 1 Report

1. Content of Introduction is short. Research background is missing.
2. What is the new contribution in your work
3. State research gap & explain reason for doing this research work
4. What is the ultimate goal of your work
5. Give citation wherever required
6. What is the limitations of your study?
7. Describe the innovation and goals of this research to differentiate your work from others.
8. Kindly add the governing equations involved in your work.
9. Have you compare your work with the other researchers??

Major revision

Reviewer 2 Report

Review of the manuscript “AF-SRNet: Quantitative Precipitation Forecasting Model Based on Attention Fusion Mechanism and Residual Spatiotemporal Feature Extraction” by Liangchao Geng et al.

The authors propose a new method using radar and rainfall data for a short-term, high-resolution, nowcasting convective precipitation event. They describe the method in detail, including a case study. Therefore, I think that the manuscript is within the scope of the journal, and it is suitable for publication after grammatical changes (below found, suggesting modifications).

Comments

Line 2. Delete “the”

Line 3. Use “research” instead of “researches”

Line 4. Use “in” instead of “on”

Line 5. Use “spatiotemporal”

Line 6. “an” instead of “a”

Line 7. “a more”

Line 9. Use “;” instead of “,”

Line 10. Use “;” instead of “,”

Line 11. Two instead of “2”

Line 19. for preventing flood disasters and reducing socioeconomic impacts

Line 26. Forecasts instead of forecast

Line 30. Start with “;” or with capital “T” Therefore, radar-based

Lines 23-24. predicts future precipitation's intensity, distribution, movement, and evolution.

Line 35. Radio echo extrapolation technology is currently a main technical means of precipitation nowcasting.

Line 38. Delete “of”

Line 37. “;” instead of “,”

Line 39. Better if end with “.” And start with “Still, …”

Line 54. Better “some time”

Line 61. Delete “the”

Line 66. Use “taking” instead of “to take”

Line 67. “radar-echo”

Line 68. Use a dot “.” Instead of a comma

Line 73, Better “This paper proposed an Attention..”

Line 78. Change to “combines”

Line 81. Delete “the”

Lines 81-83. Text suggestion “radar-based quantitative precipitation forecasting (RQPF) has been widely used in precipitation nowcasting in recent years due to the spatiotemporal discontinuity of precipitation station data”

Line 85. better to write “control; then,” and replace “this” by these”. Write “the” after “into”

Line 86. Replace “forecast” by “forecast”

Line 8. Delete “the” before important”. Write “the” after “of”. Replace “forecast” by “forecast”

Line 93. Replace “relies” by “relying”

Line 97. Delete “the”

Lines 105. Suggesting text “the precipitation system concludes with a more complex spatiotemporal motion, and the spatial and temporal information will affect each other in these methods.”

Lines 108-111. No clear text, suggesting text “when extracting the spatiotemporal information from the precipitation system, some data can be lost because the temporal and spatial information will affect each other.”

Line 112. Delete “data”

Line 114. Delete “In order to”

Line 117. Delete “the task of”

Lines 117-118. Change to “Attention mechanisms”

Line 119. Replace “proved” by “proven”

Line 120. Replace “spationtemporal” by “spatiotemporal”

Line 121. Delete “s” in features

Lines 130-132. No clear sentence. Suggesting text “Radar Encoder extracts spatiotemporal features from Radar echo maps. The precipitation Encoder extracts spatiotemporal features of precipitation maps so that the two will not interfere in the early stage.”

Line 137. Add “the” after “use”

Lines 138-140. Suggesting text “However, the temporal and spatial data will affect each other, making it difficult to extract the complex motion features in the precipitation evolution”

Line 140. Better use “designed”

Lines 144-155. No clear text. Suggesting text with some change, consideration is after you “Each module includes a structure of residual. They can effectively utilize the previous spatiotemporal state information so that the feature extraction has a wider spatiotemporal receptive field. To further extract more efficient deep spatiotemporal features, four SRUs are typically stacked into a single encoder, as shown in Figure3(b). For SRU at time step t in layer k, the temporal features TE, the spatial features SE and spatiotemporal features STE are fed into the corresponding modules of SRU. In particular, for k > 1, these features are represented with the hidden state from the previous layer H k−1 t . The hidden state of the last time step Hk t−1 includes the spatial state S k t−1 and the temporal state T k t−1; they are fed into the temporal and spatial modules, respectively. In addition, the input of the SRU also includes the temporal residual information extracted at the time before t TRk pre and the residual spatial information before the k layer SRk pre. In particular, when k = 1, there is no residual spatial information”.

Lines ?? Paragraph between 155 and 156. Please rephrase it

Lines ??. Maybe the verb “use” is better

Lines 173-174. Rephrase to “more complex dependencies exist”

Line 175. Change from “use” to “using”

Lines 175-176. Rephrase to “two encoders to independently extract radar echo features and precipitation features”.  Delete “,” and use “.” “The”

Line 177. Add “the” before “highest”. Add a space before “Then”

Line 178. Change from “,” to “;”. Delete “such”. Use a dot “.” After “4”-

Line 181. Add “the” before “microphysics”

Line ? Change to “, obtaining”

Line 189. Change to “to”

Line 181. Add a “,” before “respectively”, change to “subsequent”

Line 208. Change to “the same as the radar”

Lines 209-211. Rephase to “we down sampled the original resolution data to a size of 1 20× 140 pixels, considering the limitations of computing power and training costs. The horizontal resolution after

Line 213. Change to “the past 20 times (2 hours) data”

Line 214. Change “,” for “;”

Line 215. Change to “as the model's input”

Line 217. Add “the” before “training”

Line 221. Change to “statistically”

Line 222. Use a dot “.” After “5”.

Lines 223-225. Rephrase to “Specifically, among these categories, rainfall above 2 mm is the lowest proportion with a percentage of 2.2%, and rainfall between 0 mm to 0.2 mm is larger than rainfall above 2 mm.”

Line 236. “we”

Line 246. Change for “following”

Line 247. Add “the” after “in” and use “;” instead of “,”

Line 248. Rephase as “no longer decreases”

Line 250. Lower “w”

Line 252. Add “and” before “the”

Line 254. Delete “firstly”

Line 255. “means” instead of “mean”

Line 256. Delete “make”

Line 257. “high-intensity”

Line 258. “an” instead of “a”

Lines 259-260. Rephrase “the PredRNN, and the SE-ResUnet as it can capture short-term”

Line 261. Add “and” before “MIM”

Line 263.  Use “in” instead of “of”

Line 264. Rephrase by “enhanced; even”

Line 272. Delete “columns”

Line 272. Rephrase to “model can predict the”

Line 273. Rephrase to “However, our model undeniably suffers”

Line 274. Delete “ , which is”

Line 275. Use “present”

Line 276. Delete “in”, add “the” before “SE-ResUnet”

Line 282. Seven

Line 283. “a lower”

Line 284. Rephrase to” other models' rest times, demonstrating our approach's effectiveness.”

Line 288. Add “the” after “of”

Line 290. Delete “chose to”

Lines 292-293. Rephrase to “The results of the AF-SRNet without attention fusion block (denoted by SRNet) and STLSTM are compared to see the impact of the spatiotemporal residues feature extracted by SRNet.”

Line 295. “feature”

Line 296. Add a comma after “information”

Line 308. “method”

Line 314. Use “Currently” instead of “At present”

Line 315. Rephrase to “suffer the problem that spatial and temporal information affect each other”

Lines 317-319. Rephrase to “It is difficult to effectively establish microphysical constraints in developing precipitation systems. To solve these problems, we explore the combination of independent spatiotemporal modeling and multimodal fusion in precipitation nowcasting.”

Line 324. Change to “interpolating”

Line 325. Change “it has” to “to have”

Line 330. Change to “affecting”

Line 332. Add “a” before “wider”

Line 338. Change to “an”

Line 340. Rephrase to “spatial and temporal domains, respectively.”

Line 343. Rephrase to “forecasting precipitation area”

Line 344. Use “for”

Figures 3, 4, and 5. Some sentences belong to the main text rather than the Figure’s capture.

Suggested text modification for the Abstract.

Short-term high-resolution quantitative precipitation forecasting is essential in providing alert information to society. At this point, radar-based quantitative precipitation forecasting has been a key and challenging task in meteorological research. However, since the Z-R relation between radar and rainfall has several parameters in different areas, and the rainfall varies with seasons, traditional methods cannot capture high-resolution spatiotemporal features. Therefore, we propose an Attention Fusion Spatiotemporal Residual Network(AF-SRNet) to forecast rainfall precisely for the weak continuity of convective precipitation. Specifically, the Spatiotemporal Residual Network is designed to extract the deep spatiotemporal features of radar echo and precipitation data. Then we combine the radar echo feature and precipitation feature as the input of the decoder through the Attention Fusion Block; after that, the decoder forecasts the rainfall for the next two hours. We train and evaluate our approaches on the historical data from Jiangsu Meteorological Observatory. The experimental results show that AF-SRNet can effectively utilize multiple inputs and provides more precise nowcasting of convective precipitation.

Reviewer 3 Report

General Comments

Due to the deficiency of the observed precipitation from meteorological stations and radar reflectivity, the Attention Fusion(AF) block to fuse radar reflectivity and station-observed precipitation firstly. The Spatiotemporal Residual Unit is introduced further to extract deep features in the spatial domain and the temporal domain. Combined these two techniques, the authors proposed a Attention Fusion Spatiotemporal Residual Network (AF-SRNet) for radar quantitative precipitation forecasting. The AF-SRNet model is showed to perform best compared with other companies, so the work is valuable to be published. However, it should be improved carefully before its publication, especially for the writing. Therefore, I am asking a major revision.

Major comments:

1. The review in the introduction section can hardly let me understand why this work is necessary for both scientific and technical purposes. These references should be summarized and discussed properly to show they advantages and disadvantages, to show the question why this work must be done to address the question.

2. The language must be improved significantly to facilitate reading. The present version used too much fussy terms or profession terms without explanation.

3. The discussion and conclusion sections can be merged together, and the conclusions should be clearly present with more details.  

Minor comments:

Abstract:

L1-3: “Short-term high-resolution quantitative precipitation forecasting plays an important role in providing alert information to the society. At this point, radar-based quantitative precipitation forecasting has been a key and challenging task in meteorological researches.” This could be dropped or revised to be much brief.

L5: “traditional methods”, may be explained briefly here to let readers know why they cannot capture high-resolution spatialtemporal features.

L18-19: references may be provided

L22: 0 2 h à 0-2 h

L25-26: “Generally, for medium-range and long-term range forecast, the numerical weather prediction(NWP)[6] models driven by physics simulation provide superior and more stable predictions”

à Generally, for short-range and medium-term range forecasts the numerical weather prediction(NWP) models provide superior predictions, but models have poor performance in nowcasting (References).”

L27: weather radars à meteorological radars

L34: “in the future” à in the coming hours ?

L35: a main technical means?

L38: the movement of between frames?

L40: inverted à retrieved?

L41: “in two steps”, this should be mentioned previously when you discuss it

L42: “the forget gate” and “recall gate” may be explain somehow

L52: GAN should be used after the full name 

L93: “relies” à regarding to 

L?: “ground precipitation grid data” à gridded precipitation observations?

L197: the 3-kilometer contour, what does it mean?

L205: bilinearly interpolating or Cressman?

L209-210: computing power and training costs à computational costs?

L210: what method has been used?

L213: determined to use à used

L214: drop “Based on this,”

L214: sequences may be groups

L218: bring?

Figure 5: is this a case? On what time? Please include these informations

L233: epoch?

Figure 6. During what period? The information should be included in the caption

L?: what is “the meteorological domain”?

Equation (7): the meanings of the terms, such as TP, FN, FP, should be introduce here, so that readers will know if the score is better when it is higher or lower.

L285: the Unet model à SE-ResUNet?

Figure 8: is this still about the case in Figure 7? 

Reviewer 4 Report

This paper is well structured. It also provides a new finding in the field of precipitation analysis. I would suggest a minor modification. The authors should provide a high resolution images for Figs 7 and 9. 

Round 2

Reviewer 1 Report

Revised version is appropriate

Reviewer 3 Report

The responses are fine, and I would like to recommend it to be accepted. Please double check the manuscript again for presentation.